# Softmax Supervision with Isotropic Normalization

## Abstract

The softmax function is widely used to train deep neural networks for multi-class classification. Despite its outstanding performance in classification tasks, the features derived from the supervision of softmax are usually sub-optimal in some scenarios where Euclidean distances apply in feature spaces. To address this issue, we propose a new loss, dubbed the isotropic loss, in the sense that the overall distribution of data points is regularized to approach the isotropic normal one. Combined with the vanilla softmax, we formalize a novel criterion called the isotropic softmax, or isomax for short, for supervised learning of deep neural networks. By virtue of the isomax, the intra-class features are penalized by the isotropic loss while inter-class distances are well kept by the original softmax loss. Moreover, the isomax loss does not require any additional modifications to the network, mini-batches or the training process. Extensive experiments on classification and clustering are performed to demonstrate the superiority and robustness of the isomax loss.

## 1 Introduction

Recent years have witnessed significant progress in image classification tasks with convolution neural networks (CNN) LeCun et al. (1998); Krizhevsky et al. (2012). For classification problems, the softmax is a suitable criterion for supervised learning since it is capable of training network parameters to generate discriminative features for hyperplanes to distinguish different classes. Due to its end-to-end characteristic, CNN is amenable to learning such that we only need to feed the network with plenty of training samples. Therefore, the softmax is the most fundamental classifier applied in architectures of deep learning.

However, there are still defects of the softmax loss[1]. Features extracted by convolution layers work best only with softmax classifier. When we apply these *feature vectors* in other tasks such as image retrieval with $k$-nearest neighbors ($k$-NN) or clustering with K-means, the results are usually sub-optimal, as shown in Figure 1. To separate different classes is the sole purpose of softmax classifier, and it does not ensure that the distances (generally Euclidean distances) within the same class are smaller than inter-class ones. In order to extract better features not only for classification powered by the softmax classifier but for other tasks using distances of feature vectors, many approaches have been proposed in the past years.

One way is adding some new loss terms to the original softmax loss. The center loss Wen et al. (2016) penalizes distances of training samples to their corresponding class centers, thus enhancing the compactness of each class. Since this loss cannot extend the distances between class centers, the result depends heavily on the center initialization of all classes. Also, a relatively small batch size will seriously affect the performance of the algorithm, because centers cannot be accurately calculated with limited examples in a batch, especially for datasets with plenty of classes. The contrastive-center loss Qi & Su (2017) combines the center loss and the contrastive loss Sun et al. (2014) together to penalize distances of samples in the same classes and enlarge inter-class distances. It works well on the CIFAR-10 classification task Krizhevsky & Hinton (2009) and the LFW verification task Huang et al. (2008). To gain good performance, however, this loss needs to carefully select data batches for training too.

---

[1]For consistency with related work, we call the softmax and its related loss for optimization the softmax loss.

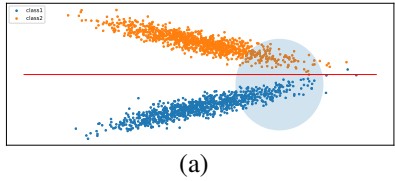 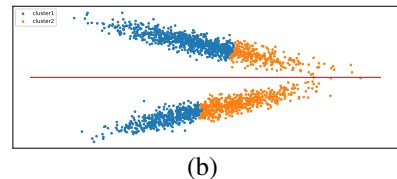

(a)                 (b)

Figure 1: (a) Features extracted by CNN with softmax loss are separated well into two classes by a line (hyperplanes in high-dimensional spaces), but lots of orange samples are included in a neighborhood (the blue circle) of a blue sample. (b) The performance of agglomerative clustering Beeferman & Berger (2000) on these features is poor.

Some other strategies aim at solving the problem in a different way. The triplet loss Schroff et al. (2015) comes up with a novel method. Instead of exploiting the softmax loss, they remove the final logits layer of network and directly minimize Euclidean distances between anchors and positive samples while maximizing distances of anchors to negative samples in triplets. However, the number of different triplets are much more than that of training samples, leading to that selecting proper triplets is crucial for the triplet loss. Or inappropriate triplets will result in slow convergence. What's more, the scalar margin in the triplet loss and the learning strategy influence the final performance of the model as well. To sum up, the principle of the triplet loss is straightforward and plausible, but the parameters and the "semi-hard triplets mining" approach make this algorithm hard to implement.

Methods mentioned above are all based on supervised optimization, meaning that they all depend on sample labels to learn discriminative features. In this paper, we propose a simple approach, named *isotropic normalization*, that reshapes data distribution towards easy classification without the aid of labels. We firstly analyze the distribution of features extracted by CNNs supervised by the softmax loss. Elliptical shapes of feature distributions lead to intra-class distances even greater than inter-class distances, indicating that the softmax loss needs to be improved for tasks using feature distances. Then we attempt to modify the feature distribution according to the global distribution itself rather than information from sample labels. Combined with the vanilla softmax, we propose a new loss, called the isotropic softmax (isomax for short) loss. For the isomax, the intra-class distances of the features can be well minimized by the isotropic loss, and at the same time, the vanilla softmax loss ensures the inter-class separability. We perform extensive experiments with different networks and different datasets to illustrate the effectiveness, simplicity and portability of our method.

## 2 ANALYSIS OF SOFTMAX

In this section, we introduce the softmax loss and analyze its limitations with visualization of 2-D features. According to the feature distribution, we show why these features are not the optimal option in tasks that use feature distances. Then we present our approach to ameliorating the feature distribution.

### 2.1 SOFTMAX LOSS

Suppose that we have $N$ training samples of $n$ classes. Let $x_i \in \mathbb{R}^d$ denote the $i$-th image feature vector and $y_i$ its label. Then softmax loss can be formulated as

$$\mathcal{L}_S = \sum_{i=1}^{N} -\log\left(\frac{\exp(f_{y_i})}{\sum_{j=1}^{n}\exp(f_j)}\right). \tag{1}$$

For typical neural networks, $f$ denotes the output of a fully connected layer with weights $W$ (a parameter matrix with $n$ columns) and bias $b$. Then the softmax loss can be rewritten as Equation 2 Liu et al. (2016)

$$\begin{aligned}
\mathcal{L}_S &= -\sum_{i=1}^{N} \log \frac{\exp(W_{y_i}^T x_i + b_{y_i})}{\sum_{j=1}^{n}\exp(W_j^T x_i + b_j)} \\
&= -\sum_{i=1}^{N} \log \frac{\exp(\|W_{y_i}\|\|x_i\|\cos(\theta_{y_i}) + b_{y_i})}{\sum_{j=1}^{n}\exp(\|W_j\|\|x_i\|\cos(\theta_j) + b_j)},
\end{aligned} \tag{2}$$

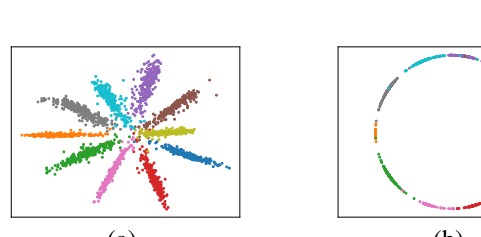 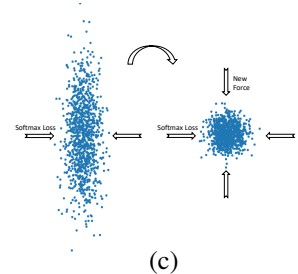

|     (a)     |     (b)     |     (c)     |

Figure 2: Visualization of MNIST. (a) Features of 1000 MNIST samples obtained with softmax loss. Different colors represent different classes. (b) Normalize 2-D features to a circle. Best viewed in color. (c) Schematic illustration of our method. We enforce a restriction on features in the approximately orthogonal direction from that of softmax to reshape the feature distribution.

| Layer | input | output | kernel | param |
|-------|-------|--------|--------|-------|
| conv1 | 28×28×3 | 28×28×32 | 5×5,1 | 2.4K |
| pool1 | 28×28×32 | 14×14×32 | 2×2,2 | |
| conv2 | 14×14×32 | 14×14×64 | 5×5,1 | 51.2K |
| pool2 | 14×14×64 | 7×7×64 | 2×2,2 | |
| fc1 | 3136 | 1024 | | 3.2M |
| drop | 1024 | 1024 | | |
| fc2 | 1024 | 2 | | 2K |

Table 1: The structure of our network for the experiment of MNIST. It is similar to LeNet LeCun et al. (1998), but we add some filters and reduce the number of neurons in the last hidden layer to 2.

where $W_j$ denotes the $j$-th column of $W$. Since $W_j$ and $x_i$ are all vectors, the inner product of them can be formulated as $\|W_j\|\|x_i\|\cos(\theta_j)$, where $\theta_j$ is the angle of $W_j$ and $x_i$. After formulating the softmax loss to an inner product form with a factor $\cos(\theta)$, it is clear that the softmax classifier is prone to enforce examples in a class have similar $\theta$ and different $\theta$ for different classes. This property of the softmax leads to the thin elliptical shape of feature distribution for each class with respect to the global center.

## 2.2 Visualization on MNIST

To analyze the feature distribution, it is a feasible way to visualize data points in a 2-D plane or 3-D space. MNIST LeCun et al. (1995) is a hand-written digit dataset of 0-9 in 10 classes, with 60,000 training samples and 10,000 testing samples. Image size of both subsets is 28×28. Since MNIST is such a simple dataset that even we reduce the dimension of feature vectors to 2, the softmax classifier is applicable to maintain 98% accuracy. So we choose it as our first example to describe the feature distribution.

The network architecture is presented in Table 1. We train the network with 60,000 training samples without data argumentation, and randomly sample 1000 testing images in 10,000 testing set. Since the dimension of feature vectors is 2, we can plot features of testing samples, as shown in Figure 2 (a). It is evident that minimizing the softmax loss leads to the separability of different classes. But for some samples, especially those near to the center of overall distribution, inter-class distances are even smaller than intra-class distances.

## 2.3 Restriction to Feature Distribution

Figure 2 (a) directly shows overall distribution under supervision of softmax loss. The thin shape of feature distribution of a certain class is the inevitable result of the softmax loss according to our previous analysis, which causes the large intra-class distances and small inter-class distances.

Apart from the softmax loss, if we can find a new "force" which can reshape the thin distribution to a "medium build" (Figure 2 (c)) , the inter-class distances will be kept by the softmax loss while the intra-class distances will be guaranteed by the new force.

For a certain class $i$, we can regard features of this class as a normal distribution $\mathcal{N}(\mu_i, \Sigma_i)$ where $\mu_i$ is the mean of this class and $\Sigma_i$ is the covariance matrix. Then there exits an orthogonal matrix $U$ and an eigenvalue-diagonal matrix $D$ satisfying $\Sigma_i = UDU^\top$. Diagonal values in $D$ can describe the shape of the normal distribution. If these eigenvalues are almost the same (with small variance), then the distribution of features is like an isotropic normal one. This analysis inspires us to devise a simple algorithm to optimize the shape of each class.

## 3 ISOTROPIC SOFTMAX

In this section, we describe our approach in detail. The relevant classification criteria are discussed as well.

### 3.1 OUR APPROACH

Inspired by Figure 2 (c) and the discussion of isotropic multivariate normal distribution, the spatial distribution of each class will be more compact if we penalize features such that $\mathcal{N}(\mu_i, \Sigma_i)$ of class $i$ will be like an isotropic one, say that the variances of different dimensions are approximately uniform.

#### 3.1.1 ISOTROPIC SOFTMAX LOSS

Here we present the isotropic loss that action on data points with the complementary effect compared with the softmax function, writing that

$$
\begin{aligned}
\mathcal{L}_I &= \frac{1}{N-1} \sum_{i=1}^{N} (\mathcal{D}_i - \bar{\mathcal{D}})^2 \\
&= \frac{1}{N-1} \sum_{i=1}^{N} \left( \|x_i - \bar{x}\|_2^2 - \frac{\sum_{i=1}^{N} \|x_i - \bar{x}\|_2^2}{N} \right)^2,
\end{aligned}
\tag{3}
$$

where $\bar{x}$ denotes the center of $x_i$, i.e. $\bar{x} = \frac{1}{N} \sum_{i=1}^{N} x_i$. In Equation 3, $\mathcal{L}_I$ denotes our new isotropic loss. In simple words, the loss we attempt to minimize is the unbiased variance estimate of distance $\|x_i - \bar{x}\|_2^2$. Specially, take $d$=2 in Figure 2 for instance. The $\mathcal{L}_I$ loss will push points near to the circle of radius $\bar{\mathcal{D}}$ and $\bar{x}$ for the circle center.

It is worth mentioning that this method is essentially different from feature normalization after training. We perform this isotropic regularizer during training and $\bar{\mathcal{D}}$ varies, meaning that the shape of each class will be deformed with softmax supervision and isotropic regularization towards easy classification during iterations.

Obviously, it is impractical and inefficient to calculate the $\mathcal{L}_I$ loss function for all training examples in each iteration. However, we can compute the estimates of $\bar{x}$ and $\bar{\mathcal{D}}$ within a batch if we randomly select samples in a training batch. The randomness of the data and the globality of the loss function guarantee the plausibility of such manipulation. Thanks to the decoupling of the isotropic loss and class labels, our loss remains unchanged from the whole training set to a batch. Details are shown in Algorithm 1.

---

**Algorithm 1** Isotropic Normalization in a Mini-Batch

---

**Input:** A batch of feature $\mathcal{B}=\{x_1, ..., x_m\}$ produced by deep neural networks
**Output:** isotropic loss $\mathcal{L}_I$ for a mini-batch
$\quad \bar{x}_\mathcal{B} \leftarrow \frac{1}{m} \sum_{i=1}^{m} x_i$               //calculate the distribution center
$\quad$ **for** $i = 1...m$ **do**              //calculate the distance to the center
$\quad\quad \mathcal{D}_i \leftarrow \|x_i - \bar{x}_\mathcal{B}\|_2^2$
$\quad$ **end for**
$\quad \bar{\mathcal{D}}_\mathcal{B} \leftarrow \frac{1}{m} \sum_{i=1}^{m} \mathcal{D}_i$           //calculate the mean distance
$\quad \mathcal{L}_I \leftarrow \frac{1}{m-1} \sum_{i=1}^{m} (\mathcal{D}_i - \bar{\mathcal{D}}_\mathcal{B})^2$
$\quad$ **return** $\mathcal{L}_I$

---

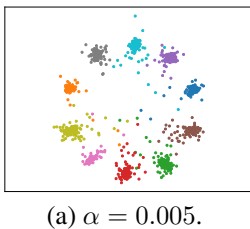 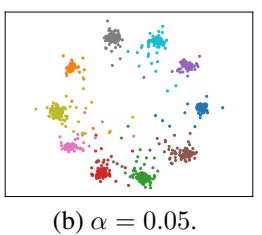 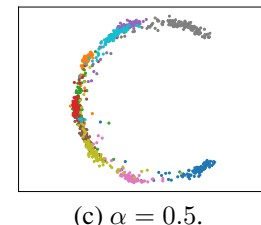

(a) $\alpha = 0.005$.        (b) $\alpha = 0.05$.        (c) $\alpha = 0.5$.

Figure 3: Feature distribution of MNIST. 10 different colors represent 10 different classes of digits 0-9. (a) and (b) Combination of isotropic loss and softmax loss with relatively small $\alpha$ supervise the network to extract better features. (c) Isotropic loss dominates the training and softmax loss loses its role due to a large $\alpha$.

The total joint loss named the isotropic softmax (isomax) loss is formulated as

$$
\begin{aligned}
\mathcal{L} &= \mathcal{L}_S + \alpha \mathcal{L}_I \\
&= -\sum_{i=1}^{m} \log \frac{\exp(W_{y_i}^T x_i + b_{y_i})}{\sum_{j=1}^{n} \exp(W_j^T x_i + b_j)} + \alpha \frac{\sum_{i=1}^{m}(\mathcal{D}_i - \bar{\mathcal{D}}_{\mathcal{B}})^2}{m - 1},
\end{aligned}
\tag{4}
$$

where $\mathcal{L}_S$ is the softmax loss and $\alpha$ controls the trade-off between the isotropic loss and the softmax loss. A small $\alpha$ may be not enough to form an isotropic distribution while a large $\alpha$ will restrain the supervision of the softmax loss, making features indistinguishable near the hypersphere. In the following section, we will discuss the influence of different $\alpha$.

As mentioned above, the loss is the unbiased variance estimate of distance $\|x_i - \bar{x}\|_2^2$, which is differentiable to $x_i$, and $x_i$ is the output of the last hidden layer of CNNs. According to chain rule, our loss term is differentiable to parameters of networks. SGD Bottou (2010), Adam Kingma & Ba (2014), RMSProp Tieleman & Hinton (2012) or other learning methods for neural networks can be used to minimize the isomax loss. As an unsupervised loss term, the optimization of the isotropic loss requires no conditions about training batches or class labels. All we need to do is combining this new loss with the original softmax loss and training the network directly via backpropagation.

### 3.1.2 BALANCE OF TWO LOSS TERMS

To find out the hyper-parameter $\alpha$, we first examine the 2-D distribution of MNIST. After 20,000 step training, the feature distribution of 1,000 test samples are shown in Figure 3. We can see that the result is much like what we expect when $\alpha$=0.005 or $\alpha$=0.05. But for a large $\alpha$, the force of the isotropic loss is so powerful that the softmax loss loses its ability to shape classes.

During training, we find that even very small $\alpha$ is able to cut down the isotropic loss quickly, the only influence of $\alpha$ is the speed of convergence. But if $\alpha > 0.1$, not only the training procedure converges slowly, but the result is also unacceptable. This threshold is suitable for almost all networks and datasets we try. In following sections, unless otherwise specified, the value of $\alpha$ is set to 0.05.

### 3.1.3 COMPARISON WITH FEATURE NORMALIZATION

Since our loss tends to normalize features to a hypersphere surface, one alternative way may be simply normalizing features to a hypersphere, specifically a circle for 2-D features (Figure 2 (b)). It needs no additional operation or loss term during training and may work well in low dimensions, but it does not mean that feature normalization will work in higher dimensions. The reason is that in high-dimensional space, the hyperplanes of the softmax classifier is much more complex. If we normalize the features to a hypersphere after training, many features from different classes would overlap together on the hyperspherical surface.

In fact, isotropic loss, as a part of isomax loss, is not a normalization operation. It only tries to gather the feature points around the sphere surface, which is a slow process. So features cannot be directly stacked together. Moreover, even if features from different classes become close to each other around the sphere surface, softmax loss, as a another supervision in isomax loss, is capable of

| Methods | Steps to converge | Label Info. | Accuracy Var. | Training Tricks |
|---|---|---|---|---|
| Center Loss | 35K steps | Yes | $\pm 0.5\%$ | Batch selection. Center initialization. |
| Triplet Loss | 100K+ steps | Yes | - | Triplet mining. Optimization method. |
| **Isotropic Loss** | 20K steps | No | $\pm 0.2\%$ | No tricks. |

Table 2: Comparison of three methods. We train these three methods for several trials on MNIST. We can see that our method converges more quickly and are more stable. Training tricks are also summarized. Fluctuation of accuracy on testing set is a measure of algorithm stability and replicability. Here only the fluctuation on softmax classification is compared (triplet loss does not have a softmax classifier). More details of performance can be found in the Experiment section.

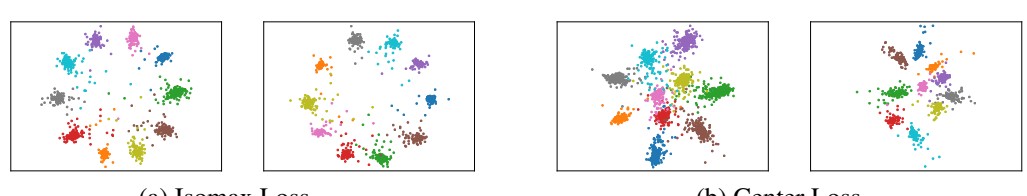

(a) Isomax Loss.                    (b) Center Loss.

Figure 4: Illustrative comparison of isomax loss and center loss. When we retrain networks with isomax loss, the overall distribution of test samples is robust and stable. For center loss, different training batches form different centers, and finally result in different global distributions.

pushing different feature points away again. Therefore our method is totally different from feature normalization after training.

## 3.2 COMPARISON WITH OTHER WORKS

There are some related works with the same purpose as ours such as the triplet loss Schroff et al. (2015) and the center loss Wen et al. (2016). We will analyze the properties of compared methods in this section. Table 2 concisely shows the merits of each method. We also discuss Batch Normalization Ioffe & Szegedy (2015) since it is an operation on feature distributions too, though for different purpose.

### 3.2.1 CENTER LOSS

Center loss is an excellent work to solve the problem of large intra-class distances. Jointly supervised by the center loss and the softmax loss, the features of same class can gather together. Equation 5 gives the formulation

$$
\begin{aligned}
\mathcal{L} &= \mathcal{L}_S + \alpha \mathcal{L}_C \\
&= -\sum_{i=1}^{m} \log \frac{\exp(W_{y_i}^T x_i + b_{y_i})}{\sum_{j=1}^{n} \exp(W_i^T x_i + b_i)} + \frac{\lambda}{2} \sum_{i=1}^{m} \|x_i - c_{y_i}\|_2^2,
\end{aligned}
\tag{5}
$$

where $c_{y_i}$ is the center of the $i$-th sample class in a batch. Centers are learned in each iteration and each batch. Though the authors of this work give some methods to avoid perturbations of centers, centers are still difficult to determine, especially when the samples from one class are limited in a batch. We train models of MNIST with the center loss and the isomax loss both twice and the experimental conditions are all the same except for the different loss terms. Results are shown in Figure 4. It is obvious that once a center is not well established, the features of this class will gather near this center, incurring that the overall distribution of associated features is unfavorable for classification. Instead the results of our isomax loss are more stable and robust. What's more, the fluctuating convergence and the additional calculation of centers make the training time longer. In our algorithm, the isotropic loss is an unsupervised loss. This loss function learns a better distribution directly from distribution itself without using labels of training data. All we need is randomly selecting samples from the training set like what we do with the softmax loss.

### 3.2.2 TRIPLET LOSS

The triplet loss is employed to supervise the learning of an Euclidean embedding per image. The loss is formulated as

$$\sum_i^N [\|f(x_i^a) - f(x_i^p)\|_2^2 - \|f(x_i^a) - f(x_i^n)\| + \alpha]_+ \tag{6}$$

where $f(x)$ is an embedding generated by CNNs, the selected anchor $x_i^a$, the positive sample $x_i^p$, and the negative sample $x_i^n$ constitute a triplet. The triplet loss directly minimizes the Euclidean distance of samples from the same class and maximizes the distance of samples from the different class until reaching a selection margin $\alpha$.

For the triplet loss, the learning strategy and the triplet selection are critical for performance. By many practices, we find that the model is sensitive to the learning rate and the learning method. Also, the converging speed is much lower than that of the model supervised by the softmax loss. Due to the massive combination of triplets and the learning tricks it requires, the results we get are not so good. To sum up, the training with the triplet loss is a laborious task.

### 3.2.3 BATCH NORMALIZATION

Batch Normalization is a method to reduce internal covariate shift in data Shimodaira (2000). They introduce $\gamma^{(k)}$ and $\beta^{(k)}$ for each activation $x^{(k)}$ in a $d$-dimensional feature $x$, and transform original feature $x=(x^{(1)}, ..., x^{(k)}, ..., x^{(d)})$ to a new feature $y=(y^{(1)}, ..., y^{(k)}, ..., y^{(d)})$ by

$$y^{(k)} = \gamma^{(k)} \frac{x^{(k)} - \mathrm{E}[x^{(k)}]}{\sqrt{\mathrm{Var}[x^{(k)}]}} + \beta^{(k)}. \tag{7}$$

By adding Batch-Norm layer (BN layer) to a certain layer in networks, each dimension of the feature of this layer is transformed according to Equation 7. Training samples in a batch are used for feature distribution transformation, which seems similar to our approach.

Though batch normalization and our method both aim at distribution transformation, they are totally different from operation and functionality. According to Equation 7, the normalization operation is along the axis of one dimension of features. Batch normalization constrains the values of each dimension of features within a reasonable range, while our method normalizes Euclidean distances of feature vectors with respect to the global center of data distribution. To make it clearer, we interpret a simplified example by letting $\gamma^{(k)}=1$ and $\beta^{(k)}=0$. As each dimension is normalized, batch normalization transforms features inside a hypersphere. Our loss term tries to push features onto the spherical surface, thus reducing the variance of distances to the distribution center (spherical center). Batch normalization prevents the training from getting stuck. As for our approach, a final feature map with small intra-class distance and large inter-class distance is our target. In fact, these two seemingly similar operations are mutually independent. If we use batch normalization for the last hidden layer, the $x$ in Algorithm 1 will be replaced by the output $y$ of batch normalization. In the following experiments in the next section, batch normalization is applied together with our new loss.

Briefly in summary, batch normalization is a local normalization on a certain feature map to improve the training of the network. Ours is a global geometric supervision on the overall feature distribution of the last hidden layer to acquire better representations.

## 4 EXPERIMENTS

We evaluate the isomax on three tasks: image classification, feature clustering and face verification. Experiments are performed on four datasets: MNIST, CIFAR-10, a subset of ILSVRC2012 Deng et al. (2012) with 200 classes, and CASIA-WebFace Yi et al. (2014). We divide CASIA-WebFace into training and testing sets with a ratio of 2:1. These four datasets are very different in data attributes, data volumes, and class numbers. For the classification task, the testing sets of these four datasets are all used. For the clustering task, the Face Recognition Grand Challenge (FRGC) Phillips et al. (2005) dataset is also used for testing. For face verification, we use the Labeled Faces in the

Wild (LFW) benchmark dataset. To compare fairly for each dataset, we train three models under different supervision: the softmax loss, the center-softmax joint loss and our isomax loss using the same training methodology and network architecture. For verification task, we also train a model with the triplet loss since it achieves state-of-the-art performance in LFW. All the experiments are implemented with TensorFlow Abadi et al. (2016).

## 4.1 EXPERIMENT SETTINGS

### 4.1.1 MNIST AND CIFAR-10

For MNIST, the network we use is shown in Table 1. For CIFAR-10, inspired by the architecture of VGG-net Simonyan & Zisserman (2014), we design our network in this way: 3 cascaded convolution blocks followed by a dropout Hinton et al. (2012) layer and then a fully-connected layer . There are 4 cascaded convolution layers with size of 3×3 (stride=1) and a max-pooling layer with size of 2×2 (stride=2) in each block. Filter numbers in blocks are 64, 96 and 128. We do not use data augmentation on these two datasets but use a dropout of 0.8. The initial learning rate is 0.1 and decays with an exponential rate of 0.96 every 1000 steps. The Adagrad method is employed for optimization. Batch size is 128.

### 4.1.2 ILSVRC-SUB

For this dataset, there are 200 classes and about 1300 images in each class. The Inception-ResNet-v1 network Szegedy et al. (2017) is applied. We do not use the input size of 299×299 according to the original paper, but 160×160 for simplicity of calculation. In training phase, we resize the short edge of image to 182, and randomly crop a 160×160 window. The random adjustment of hue, brightness, contrast and saturation is applied. In testing phase, we resize the short edge to 160 and crop a square sub-image in the center. We harness Adagrad and a decaying learning rate that starts at 0.05 and decays with an exponential rate of 0.94 every two epochs. Batch size is 128.

### 4.1.3 CASIA-WEBFACE

CASIA-WebFace contains 0.49M labeled face images from over 10,575 individuals. We firstly exploit MTCNN Zhang et al. (2016) to align face images based on 5 points. Then two-thirds of images from each individual (about 0.32M totally) are randomly selected to train the Inception-ResNet-v1 network. During training, we resize images to 182×182 and randomly crop a 160×160 window, while for testing we directly resize images to 160×160. Random left-to-right flipping is also used for training. The training method we use is RMSProp with decay of 0.9 and $\epsilon = 1.0$. Batch size is 512. In fact, this relatively large batch size is chosen for center loss since centers need to be updated in a batch and the number of classes of CASIA dataset is too large. We use an initial learning rate of 0.1, divided by 10 after 50 epochs and 65 epochs.

## 4.2 CLASSIFICATION

The softmax loss is our baseline, which is designed for image classification. So we firstly evaluate our model on this kind of task. To ensure that our modification to the original softmax loss has no negative impact on the softmax classifier, we firstly evaluate the accuracy on four datasets. Then, to illustrate the superiority of our model in distance-based tasks, we also report the $k$-NN classification results using feature vectors attained by the supervision of compared losses. Table 3 shows the experimental results.

We observe that our algorithm does not deteriorate the performance of the softmax classifier, while in $k$-NN classification tasks, our proposed algorithm performs better, especially for datasets with many classes. For these datasets, it is more essential to place features of different classes in a finite feature space properly. To show details of training, we take the CASIA-WebFace dataset as an example, plotting the curve of accuracy during training process in Figure 5.

Another interesting evaluation we analyze here is shown in Figure 6. In three contrast experiments, we monitor the value of center loss, which is only minimized as a loss term in the experiment of center loss. However, for each dataset, the isomax loss makes the center loss much lower, though we do not take the initiative to optimize it. As the center loss is a measure of intra-class distance,

| Datasets | Methods | Softmax Acc (%) | $k$-NN Acc (%) | | |
|---|---|---|---|---|---|
| | | | $k$=1 | $k$=5 | $k$=10 |
| MNIST | Softmax | 98.75 | 97.50 | 97.81 | 97.73 |
| | Center Loss | 98.44 | 98.03 | 98.20 | 98.05 |
| | **Isomax** | **98.60** | **98.20** | **98.63** | **98.28** |
| CIFAR-10 | Softmax | 90.45 | 83.75 | 87.42 | 87.89 |
| | Center Loss | 90.61 | 84.90 | 88.05 | 88.75 |
| | **Isomax** | **90.30** | **85.39** | **89.14** | **89.53** |
| ILSVRC-sub | Softmax | 75.89 | 47.50 | 49.55 | 52.63 |
| | Center Loss | 75.60 | 48.01 | 50.73 | 53.57 |
| | **Isomax** | 75.65 | **49.92** | **55.68** | **57.88** |
| CASIA-WebFace | Softmax | 85.87 | 48.17 | 33.94 | 28.60 |
| | Center Loss | 85.76 | 48.55 | 34.24 | 28.73 |
| | **Isomax** | **85.88** | **50.92** | **38.23** | **34.12** |

Table 3: Classification results.

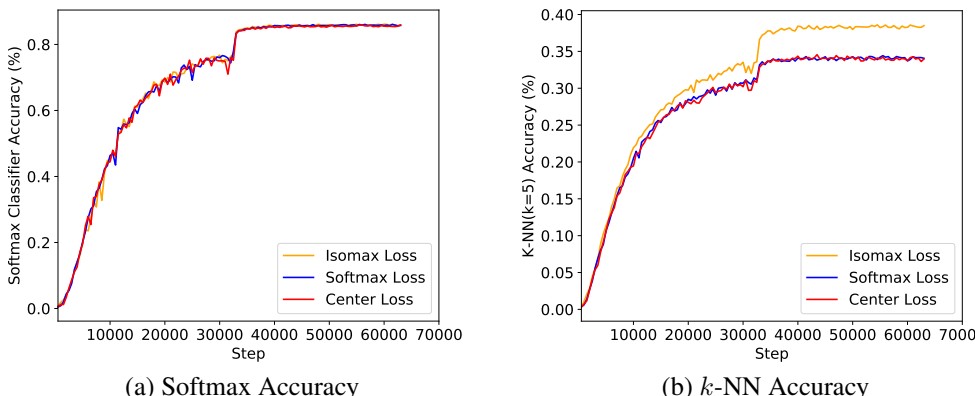

(a) Softmax Accuracy          (b) $k$-NN Accuracy

Figure 5: Accuracy curves of CASIA-WebFace testing set. The boost at 32K step is because of the decay of learning rate. (a): Softmax classifier accuracy relative to training steps. (b): $k$-NN ($k$=5) classification accuracy.

this proves that, by reshaping the overall global distribution, our algorithm can reduce the distance within a class more effectively.

## 4.3 ADVERSARIAL ROBUSTNESS

In this section, we test the performance of different losses when associated neural networks are attacked by adversarial samples[2]. To do so, we employ the Fast Gradient Sign Method to generate the adversarial samples Dong et al. (2017), say

$$\text{adversarial} = \text{original} + \epsilon * \text{sign}(\nabla_x J(x, y)), \tag{8}$$

where $\epsilon$ controls the degree of adversary. Small epsilon means slight perturbation and weak attacking. The gradient in the above formula is derived from the derivative of the softmax loss for each of three compared models.

The experiments are performed on MNIST and CAFAR-10. The experimental procedures are kept the same with ones in section 4.2. From the results shown in Table 4, we can see that the isomax loss consistently outperforms the other two losses. It is worthing noting that the superiority of the isomax loss is significant for $\epsilon = 0.003$ on MNIST. The underlying reason is presumably that classes are still maintained separable due to the more compact isotropic distribution for the isomax loss under the slight perturbation whereas the distribution of classes for the softmax and center losses might be

---

[2]Note that our algorithm is not developed for defending adversarial samples. We only use them to test the robustness of neural networks under the same degree of adversarial attacking.

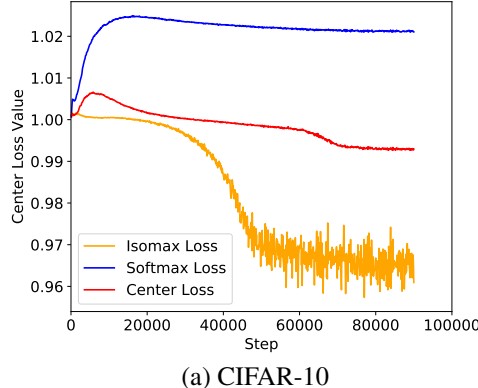

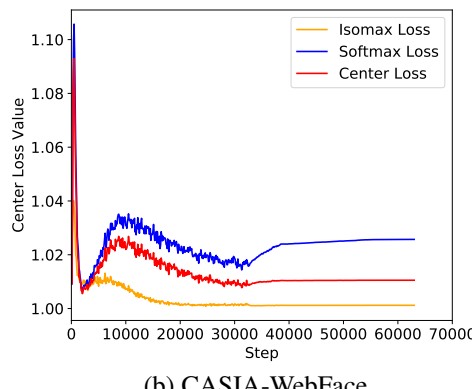

(a) CIFAR-10 (b) CASIA-WebFace

Figure 6: Values of center loss relative to training steps on different datasets. Two of four datasets are shown here as examples.

| Methods | $\epsilon$=0.003 | $\epsilon$=0.01 | $\epsilon$=0.03 |
|---|---|---|---|
| Softmax | 76.43% | 49.28% | 25.39% |
| Center Loss | 77.56% | 49.65% | 26.01% |
| **Isomax Loss** | **85.20%** | **54.57%** | **27.83%** |

| Methods | $\epsilon$=0.003 | $\epsilon$=0.01 | $\epsilon$=0.03 |
|---|---|---|---|
| Softmax | 70.12% | 39.84% | 20.93% |
| Center Loss | 69.53% | 37.68% | 19.38% |
| **Isomax Loss** | **74.34%** | **43.75%** | **22.32%** |

(a) MNIST (b) CIFAR-10

Table 4: Classification accuracy on MNIST and CIFAR-10 for adversarial attacking. The adversarial samples are produced with the fast gradient sign method (FGSM) with different $\epsilon$.

messy, especially near boundaries. To our surprise, however, the center loss performs the worst on CIFAR-10, implying its weak robustness under adversarial perturbation.

## 4.4 CLUSTERING

Compact intra-class features can be used in clustering tasks. We evaluate K-means and agglomerative clustering (ward linkage) performance of features under different supervision signals. For MNIST, CIFAR-10, and ILSVRC-sub, 10,000 images from their corresponding testing set are used for clustering. As for the face model, we use 12,776 face images of 466 different identities from FRGC Phillips et al. (2005) dataset. For evaluation, 12 pre-trained models mentioned above are employed, i.e. 3 different losses for each of 4 datasets. Normalized Mutual Information (NMI) Strehl & Ghosh (2002) is selected as the clustering performance evaluation metric. The results in Table 5 show the consistent superiority of our algorithm.

## 4.5 FACE VERIFICATION

We also evaluate the performance of our algorithm on a widely used verification benchmark LFW Huang et al. (2008). The dataset contains 13,233 faces from 5749 individuals with different poses and expressions. We use the models trained on 0.32M CASIA-WebFace mentioned above. We also implement a model trained with triplet loss with the same training data and network. There is no overlap between our training data and LFW dataset. We use MTCNN Zhang et al. (2016) to align the face images of LFW like what we do on CASIA-WebFace. Instructed by the protocol for unrestricted with labeled outside data Huang & Learned-Miller (2014), we report the result of verification performance of 6,000 pairs of faces in Table 6 and Figure 7. We only train our model with 0.32M outside face images on a *single network*, so our purpose is not to improve state-of-the-art accuracy but a good comparison of isomax loss, center loss, softmax loss and triplet loss under the same condition.

Our proposed method is more efficient and suitable in verification task compared with softmax loss baseline and also outperforms center loss and triplet loss under the same condition. Since the dataset we use for training has more than 10,000 classes, making centers for each class difficult to determine

| Datasets | Methods | K-means NMI score | Agglomerative NMI score |
|---|---|---|---|
| MNIST | Softmax | 0.9031 | 0.9110 |
| | Center Loss | 0.9270 | 0.9098 |
| | **Isomax** | **0.9666** | **0.9404** |
| CIFAR-10 | Softmax | 0.7366 | 0.7136 |
| | Center Loss | 0.7428 | 0.7243 |
| | | **0.7837** | **0.7541** |
| ILSVRC-sub | Softmax | 0.7191 | 0.7658 |
| | Center Loss | 0.7193 | 0.7713 |
| | **Isomax** | **0.7349** | **0.7831** |
| CASIA-WebFace | Softmax | 0.9404 | 0.9685 |
| | Center Loss | 0.9421 | 0.9691 |
| | **Isomax** | **0.9461** | **0.9721** |

Table 5: Clustering results.

| Methods | Networks | Outside Data | Acc(%) |
|---|---|---|---|
| Yi et al. (2014) | 1 | 0.49 M | 97.73 |
| DeepFace Taigman et al. (2014) | 3 | 4 M | 97.35 |
| FaceNet Schroff et al. (2015) | 1 | 200 M | 99.63 |
| Deep FR Parkhi et al. (2015) | 1 | 2.6 M | 98.95 |
| DeepID-2+ Sun et al. (2015) | 25 | - | 99.47 |
| Softmax Loss | 1 | 0.32 M | 97.50 |
| Center Loss | 1 | 0.32 M | 97.63 |
| Triplet Loss | 1 | 0.32 M | 96.88 |
| **Isomax Loss** | **1** | **0.32 M** | **98.03** |

Table 6: Verification accuracy on LFW.

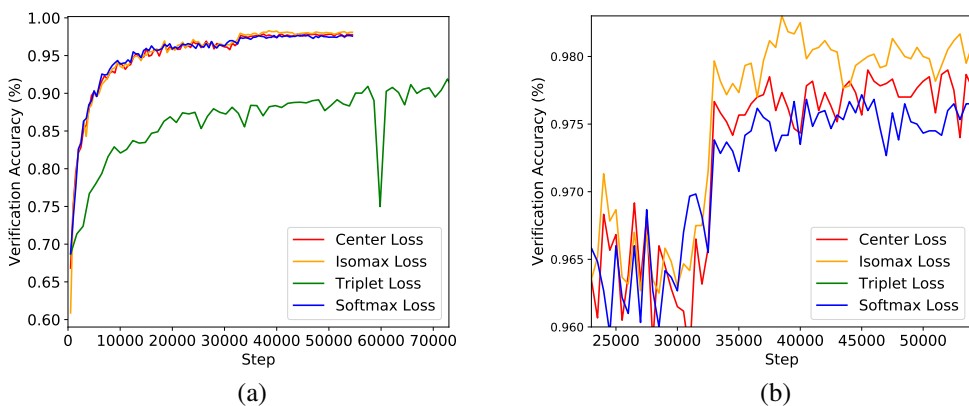

(a)                               (b)

Figure 7: LFW verification accuracy during training. (a): Triplet loss converges very slowly. (b): Zooming in the final stage of training shows that our proposed loss achieves better performance.

in a mini-batch. So the model with center loss surpasses the model with softmax loss by a small margin while ours outperforms it by a relatively large margin. Triplet loss is another baseline we want to compare with. Due to its long training process, we partially plot it in Figure 7. The accuracy of triplet loss finally plateaus at 96.88%, which is lower than the accuracy reported in the original paper. The small amount of training data and complex training tricks that we do not tune well may be the reasons.

## 5 CONCLUSION

In this paper, we propose a new isotropic loss together with the original softmax loss named isomax loss. With the joint supervision signal, CNNs generate more isotropic feature distribution for each class, and the Euclidean distance in the class decreases. The 2-D visualization and extensive experiments on different datasets for different tasks show the effectiveness of our approach. Comparison to other related works illustrates the advantage of our method on tasks using feature distance.

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
