# OpenReview forum: "Softmax Supervision with Isotropic Normalization"
_ICLR.cc/2018/Conference — Reject_

### Official Review · AnonReviewer2 · 2017-11-26
**A solid empirical study on an isotropic  softmax loss function**

**Rating:** 6
**Confidence:** 3

**Review:**

The paper studies the problem of DNN loss function design for reducing intra-class variance in the output feature space. The key contribution is proposing an isotropic variant of the softmax loss that can balance the accuracy of classification and compactness of individual class. The proposed loss has been compared extensively against a number of closely related approaches in methodology. Numerical results on benchmark datasets show some improvement of the proposed loss over softmax loss and center loss (Wen et al., 2016), when applied to distance-based classifiers such as k-NN and k-means.

Pros:

- The idea of isotropic normalization for enhancing compactness of class is well motivated

- The paper is mostly clearly organized and presented.

- Numerical study shows some promise of the proposed method.

Cons:

-  The novelty of method is mostly incremental given the prior work of (Wen et al., 2016) which has provided a slightly different isotropic variant of softmax loss.

- The training procedure of the proposed method remains unclear in this paper.

---

> ### Author Response · Authors · 2017-12-06
> **Thanks for your review**
>
> Thanks for your review.
> Given the cons you noted, I want to make further explanation. Our work does have the same purpose as Wen et al., 2016. However, from the principle to the training process, our method is completely different from the center loss. In fact, solving pitfalls of center loss is one of the motivation of our work. As for the training procedure, due to that our isotropic loss function does not require sample labels, the training procedure is almost the same as traditional softmax. The only thing we need to do is replacing the softmax loss with the joint loss. So in the paper, we did not elaborate on the training procedure.

---

### Official Review · AnonReviewer1 · 2017-11-28
**This paper presented a modification to the Center loss softmax for feature learning**

**Rating:** 4
**Confidence:** 5

**Review:**

In the centre loss, the centre is learned. Now it's calculated as the average of the last layer's features
To enable training with SGD, the authors calculate the centre within a mini batch

---

> ### Author Response · Authors · 2017-12-06
> **Thanks for your review**
>
> Thanks for your review.
> Our work is not a "modification" to center loss, but a completely new approach from concept to training process. In fact, the only similar thing we share with center loss is the motivation of facilitating the supervision capability of softmax for different tasks. That's why we focus on comparing these two methods in our paper. Specifically, we have the following key differences with center loss:
> 1) Center loss calculate centers for each class. Our algorithm calculates the global center of all samples. This calculation is one step in our Isotropic Normalization. Our isotropic loss is a measure of the variance of the distance from feature points to the global center.
> 2) Center loss is a direct way to reduce the distance within a class, which cannot enlarge the inter-class distance. Our algorithm reduces the intra-class distance indirectly by reducing the variance of feature points to the global center distance (Isotropic Normalization), and inter-class distance is also maintained by original softmax.
> 3) Sufficient experiments show that the training process of our algorithm is more simple and stable and the final results are better.

---

### Official Review · AnonReviewer3 · 2017-12-04
**simple tweak to softmax**

**Rating:** 4
**Confidence:** 4

**Review:**

After the rebuttal:

I do not think I had a major misunderstanding of the paper. I was aware that the features mostly refers to the inputs to softmax. In my point 4, I was suggesting that in order to have clustering performance, one might alternatively work on the softmax outputs instead of the inputs.

My opinion on this paper remains, and I think the contribution of this paper to machine learning is not very clearer at the current stage. It might be the case that the considered scenarios indeed happen in computer vision related problems, but I am not an expert in that regard.

========================================================================

This paper proposes a regularization to the softmax layer, which try to make the distribution of feature representation (inputs fed to the softmax layer) more meaningful according to the Euclidean distance. The proposed isotropic loss in equation 3 tries to equalize the squared distances from each point to the mean, so the features are encouraged to lie close to a sphere. Overall, the proposed method is a relatively simple tweak to softmax. The authors show that empirically, features learned under softmax loss + isotropic regularization outperforms other features in Euclidean metric-based tasks.

My main concern with this paper is the motivation: what are the practical scenarios in which one would want to used proposed method?
1. It is true that features learned with the pure softmax loss may not presents the ideal  similarity under the  Euclidean metric (e.g. the problem depicted in Figure 1),  because they are not trained to do so: their purpose is just to predict the correct label.  While the proposed regularization does lead to a nicer Euclidean geometry, there is not sufficient motivation and evidence showing this regularization improves classification accuracy.

2. In table 2, the authors seem to indicate that not using the label information in the definition of Isotropic loss is an advantage. But this does not matter since you already use the labels in the softmax loss.

3. I can not easily think of scenarios in which, we would like to perform KNN in the feature space (Table 3) after training a softmax layer. In fact, Table 3 shows KNN is almost always worse than softmax in terms of classification accuracy.

4. Running kmeans or agglomerative clustering in the feature space (Table 5) *using the Euclidean metric* is again ill-posed, because the softmax layer is not trained to do this. If one really wants good clustering performance, one shall always try to learn a good metric, or , why do not you perform clustering on the softmax output (a probability vector?)

5.  The experiments on adversarial robustness and face verification seems more interesting to me, but the tasks were not carefully explained for someone not familiar with that literature. Perhaps for these tasks, multi-class classification is not the most correct objective, and maybe the proposed regularization can help, but the motivations are not given.

---

> ### Author Response · Authors · 2017-12-06
> **Thanks for your review**
>
> Thanks for your review.
> 1) First, I want to point out that you misunderstood a fundamental aspect of our algorithm. The feature vector we described in the paper is not the softmax output (a probability vector). Instead, it is the feature vector fed to softmax, i.e. the feature vector produced by fully connected layer. Such feature vectors yielded by CNN trained with softmax supervision have far more usage than classification tasks in computer vision and speech recognition.
>
> 2) We can train different CNNs for different tasks, such as classification and clustering. However, we prefer one framework to handle more tasks in some scenarios where only limited computation and memory are available, mobile phones and surveillance devices for example. In modern intelligent phones, there are two must-need tasks for album: recognizing persons (classification) and automatically organizing photos (clustering). For phone cameras, there are also scene classification and photo editing when capturing photos. For efficiency of manipulation, the hardware of phones cannot afford many different models for many different tasks. So, we pursue one framework for many tasks. Actually, it is analogous for many embedded devices. This is the necessity of our algorithm.
>
> 3) As for the other datasets we used in experiments, we are just trying to make sure that the improved softmax does get the feature distribution we want, and make a fair comparison with other state-of-the-art algorithms like Center Loss or Triplet Loss. The result of clustering and KNN directly illustrate that our algorithm can make features more discriminative.
> We emphasize that our algorithm does not require label information because it is indeed one of the key reason why our algorithm is more simple and stable in practice. For example, If you have 10000 classes and your batch size is 128, for many classes there is only 1 or 2 samples in a batch. It is impossible to calculate the center for a class accurately. This reason leads to slow divergence and fluctuation of training with center loss. Since our algorithm doesn't need labels, we do not need to be aware of any additional considerations when building a batch or training on different datasets.

---

### Decision · Program_Chairs · 2018-01-29
**ICLR 2018 Conference Acceptance Decision**

**Decision:**

Reject

**Comment:**

The paper proposes a small modification to the loss on a network's softmax layer, which encourages the input to the softmax to be isotropic for each class.  This is motivated by applications where one may want to measure Euclidean distance between the feature vectors.  The main concern is that it's not clear that there is really a widespread need for this modified loss, i.e. that there are many applications (or a few very important ones) where one would want to train a network with a softmax layer but also need the Euclidean distances in the feature space to be sensible.